# Methods Combining Genomic and Epidemiological Data in the Reconstruction of Transmission Trees: A Systematic Review

**DOI:** 10.3390/pathogens11020252

**Published:** 2022-02-15

**Authors:** Hélène Duault, Benoit Durand, Laetitia Canini

**Affiliations:** 1Epidemiology Unit, Paris-Est University, Laboratory for Animal Health, French Agency for Food, Environment and Occupational Health and Safety (ANSES), 94700 Maisons-Alfort, France; helene.duault@anses.fr (H.D.); benoit.durand@anses.fr (B.D.); 2Faculté de Médecine, Université Paris-Saclay, 94270 Le Kremlin-Bicêtre, France

**Keywords:** transmission tree, genomic epidemiology, who-infected-whom

## Abstract

In order to better understand transmission dynamics and appropriately target control and preventive measures, studies have aimed to identify who-infected-whom in actual outbreaks. Numerous reconstruction methods exist, each with their own assumptions, types of data, and inference strategy. Thus, selecting a method can be difficult. Following PRISMA guidelines, we systematically reviewed the literature for methods combing epidemiological and genomic data in transmission tree reconstruction. We identified 22 methods from the 41 selected articles. We defined three families according to how genomic data was handled: a non-phylogenetic family, a sequential phylogenetic family, and a simultaneous phylogenetic family. We discussed methods according to the data needed as well as the underlying sequence mutation, within-host evolution, transmission, and case observation. In the non-phylogenetic family consisting of eight methods, pairwise genetic distances were estimated. In the phylogenetic families, transmission trees were inferred from phylogenetic trees either simultaneously (nine methods) or sequentially (five methods). While a majority of methods (17/22) modeled the transmission process, few (8/22) took into account imperfect case detection. Within-host evolution was generally (7/8) modeled as a coalescent process. These practical and theoretical considerations were highlighted in order to help select the appropriate method for an outbreak.

## 1. Introduction

Understanding transmission dynamics is pivotal in controlling and preventing infectious diseases. Studies have aimed to reconstruct transmission trees depicting transmission histories of actual outbreaks in order to draw conclusions on how the disease spread [1,2]. For instance, transmission trees have been used to explore hypotheses on mechanisms of transmission [3] and evaluate key transmission parameters, such as the reproduction number R, that is, the number of secondary cases caused by one infected individual [4]. In a transmission tree, nodes represent infected hosts (i.e., entities that the pathogen can infect, e.g., individuals or groups of individuals like farms in a foot-and-mouth disease (FMD) outbreak [5]), connected by transmission events represented by directed edges [6]. Transmission events in a transmission tree generally correspond to the first infection of each host, as superinfections (infection with an additional strain) or reinfections (second infection after clearance) are usually disregarded.

One way to infer transmission history in an outbreak has been the use of contact tracing methods, in which infected individuals are interrogated regarding time of symptom onset, duration of disease, and potential exposures to pathogen [7]. However, data collected by epidemiological investigations are not always available, reliable, or detailed enough for accurate reconstructions [8]. In addition, the fact that not all infected individuals are known hinders the reconstruction of an observed outbreak. Indeed, asymptomatic infected individuals are less likely to be detected unless a testing strategy has been implemented, and even then, test sensitivity (Se) and field conditions are sometimes mediocre. For instance, on-the-field implementation of the intradermal tuberculin skin test for bovine tuberculosis can differ from the official guidelines (e.g., not respecting the recommended injection area, qualitative reading of results), which in turn lowers the Se [9].

Complementary to epidemiological data, pathogen isolation and subsequent partial or total sequencing of pathogen genomes can inform on the relative closeness of strains. The increasing availability and affordability of sequencing contributes to its mounting popularity and its frequent use in molecular epidemiology. Genomic data have been frequently applied to the reconstruction of phylogenetic trees, which describe the evolutionary relationships between sequences [10]. Indeed, numerous methods and tools exist to reconstruct phylogenetic trees (e.g., [11,12,13,14]). Some studies have considered phylogenetic trees to be partially observed transmission trees [15]. However, others have highlighted the differences between these two notions [5,16,17,18]. Contrary to a transmission tree, internal nodes in a phylogenetic tree represent hypothetical common ancestors and tips correspond to sampled sequences, therefore ancestries between sampled sequences cannot be recovered from a phylogenetic tree on its own [16]. Moreover, nodes are linked by branches, which represent genetic distances, and the timing of nodes correspond to within-host diversification events (reconstructed as coalescent events), which precede transmission when considering a complete bottleneck [5,18]. A complete bottleneck means that during infection, only one strain can be transmitted, as opposed to a weak transmission bottleneck that allows multiple strains to be transmitted. Thus, without explicitly representing the hosts in which each pathogen lineage was present, we cannot identify and time transmission events from a phylogenetic tree. Phylogenetic tree reconstruction has been used to identify “transmission clusters”, that is, clusters of sequences more closely related in the evolutionary process and therefore considered epidemiologically linked. For instance, a review on HIV “transmission clusters” definitions showed that a majority were based on statistical criteria defining how likely the existence of the node was (phylogenetic node support) or a combination of phylogenetic node support and a genetic distance threshold [19].

However, inferring actual transmission trees solely from genetic data proves challenging. Indeed, genetic diversity between sampled sequences hinges on the evolutionary rate of the pathogen as well as time-to-sampling, and when diversity is limited, the ability to infer correct transmission histories is affected [20]. For example, in a *Mycobacterium bovis* outbreak, a high proportion of sampled sequences isolated from different hosts can be identical [21] due to the low evolutionary rate. While sequenced strains from pathogens that tend to have a high evolutionary rate show greater dissimilarity, a non-negligible within-host diversity and/or a weak bottleneck complicates the inference of the transmission tree solely from genetic data [22]. Thus, methods were developed to combine epidemiological and genomic data, whether in a simultaneous [5,23,24] or sequential (integrating one type of data then the other, e.g., [2,17]) manner to infer possible transmission trees.

According to graph theory, the number of spanning trees that can be constructed from a complete graph of *n* nodes is given by Cayley’s tree formula: *n^n^*^−2^ [25]. When applied to transmission trees, this number corresponds to the number of unrooted transmission trees compatible with *n* hosts. For instance, when considering 10 hosts, 10^8^ transmissions trees are compatible. Therefore, simply enumerating all possible oriented transmission trees is not a viable option when *n* is high and other strategies need to be applied. Methods that combine both epidemiological and genomic data can model four unobserved processes mentioned by Klinkenberg et al. (2017) [26] that can be defined as follows:Mutation: includes nucleotide “indel” (either a deletion or an insertion, i.e., a nucleotide disappears from or is added to the sequence) and substitution (a nucleotide in the sequence changes into another).Within-host evolution: represents how the pathogen genome changes within an individual or a group of individuals, which leads to genome diversification.Transmission: passage of a pathogen from an infected host to a susceptible host and the subsequent infection in the newly infected host. In transmission models, assumptions are thus made regarding how the disease was introduced in the host population then spread from host to host, as well as regarding the natural history of the disease.Case observation: process of identifying and sampling infected hosts in the host population.

We systematically reviewed the literature for methods combining genomic and epidemiological data to reconstruct transmission trees. A problem arises from the existence of numerous methods: how to select the appropriate method for the studied outbreak. Therefore, our goal was to discuss methods according to the epidemiological and genomic data necessary to implement them, as well as the underlying sequence mutation, within-host evolution, transmission, and case observation models.

## 2. Results

After removal of duplicates, 496 articles were imported to EndNote and screened for their relevance to transmission tree reconstruction methodology. Among these 496 articles, the full texts of 98 articles were screened for eligibility (Figure 1). The reasons for exclusion of full-text articles are detailed in Appendix A. The main reasons were as follows: the article did not actually aim to infer a transmission tree according to our definition (*n* = 23), the kind of genetic data considered (*n* = 12), or the lack of a formal combination of the two types of data (*n* = 21).

Twenty-two different methods were used in the remaining 41 articles, and we defined three families: a non-phylogenetic family, a sequential phylogenetic family, and a simultaneous phylogenetic family. In the non-phylogenetic family (NPF), phylogenetic trees were not considered in the transmission tree reconstruction, and instead, pairwise genetic distances were estimated. In the phylogenetic families (PF), transmission trees were inferred from phylogenetic trees either by using the phylogenetic tree as a source of information or by establishing a method to link the two types of trees, that is, a transmission tree was obtained by inferring the host of each node or branch in the phylogenetic tree. In the sequential phylogenetic family (SeqPF), phylogenetic trees had to be reconstructed prior to the implementation of the transmission tree reconstruction methods. However, in the simultaneous phylogenetic family (SimPF), phylogenetic and transmission trees were simultaneously inferred. We decided to distinguish between the two since the two-step approach in the sequential phylogenetic family means the users need to choose an appropriate method to reconstruct the phylogenetic tree and implement it, prior to the transmission tree reconstruction method. Thus, the sequential phylogenetic family assumes that the phylogenetic tree does not depend on the transmission process.

To illustrate the problem these three families tried to address, Figure 2A shows a simplistic transmission and within-host evolution scenario: D transmits to U (an unobserved individual), who in turn transmits to C and A, and finally, C transmits to B. In this figure, the length of each host rectangle represents the time from infection to removal (either recovery or death). From this small outbreak, we consider the sequences a, b, c, and d collected respectively from hosts A, B, C, and D at times T_A_, T_B_, T_C_, and T_D_. The removal times of known hosts are also included in the data: R_A_, R_B_, R_C_, and R_D_. From the known epidemiological data and either pairwise genetic distances (NPF) or the phylogenetic tree (Figure 2B, PF), each family of methods aimed to reconstruct the transmission tree (Figure 2C), with or without inferring the unknown transmission times t [infector, infected].

Table 1 presents the epidemiological and genomic data needed to implement each method. A majority (20/22) of methods used at least sampling times (Table 1). Eleven methods considered removal times, seven the onset of infectiousness, while few (3/22) considered the start of exposure (Table 1). Moreover, intrinsic characteristics (predominant species, number of animals, production period) are only considered in two methods, belonging to the NPF and SimPF, respectively: Aldrin 2011 [28] and BORIS (Bayesian Outbreak Reconstruction Inference and Simulation) [29] (Table 1). Similarly, only two other methods included contact data in their transmission model: one in the NPF, outbreaker2 [30], and one in the SeqPF, TiTUS [31] (Table 1). Ten out of the twenty-two methods (Table 1) were implemented in packages, 7 available as R packages (however, since their implementation, two have been removed from the CRAN repository; for details see Appendix A), one code on github (Transmission Tree Uniform Sampler, TiTUS), and the remaining two on BEAST [13] (beastlier) or BEAST2 [14] (Structured COalescent Transmission Tree Inference, SCOTTI).

Within-host evolution was explicitly modeled in fewer than half of the methods (8/22), and most methods made restricting assumptions on the outbreak: all cases are observed and sampled, the transmission bottleneck is complete, or a single introduction event took place (Table 2, Table 3 and Table 4). Observation is the detection of an infected host, and a host is sampled when a pathogen sequence was isolated.

### 2.1. Non-Phylogenetic Family

The non-phylogenetic family (Figure 3) contained eight methods. The majority of these methods (5/8) attached a genetic model that described the pairwise genetic distance between two individuals according to their relationship in the transmission tree to an explicit model of disease transmission. In the Bayesian methods (4/5), these models were combined in a likelihood function, which was used to sample from the transmission tree space.

#### 2.1.1. Methods That Consider Mutations to Occur at Transmission

The Bayesian method proposed by Ypma et al. (2012) used three types of data (temporal, geographical, and genetic) from an H7N7 outbreak in poultry farms in the Netherlands and considered them all independent of each other. The likelihood function was therefore a product of contributions given by the three types of data [32]. Similarly, Jombart et al. (2014) decomposed the likelihood into a genetic likelihood and a temporal likelihood in the outbreaker package [24]. Campbell et al. (2019) then extended the transmission model in this method to include contact data in a reporting likelihood in outbreaker2 [30]. Probability of transmission between two sampled individuals was inferred from known generation time Tg and time-to-sampling distributions. In addition, outbreaker and outbreaker2 considered two parameters to model unobserved cases: π, the proportion of sampled cases in the outbreak, and κ, the maximum number of generations separating a sampled infected individual and his sampled ancestor in the transmission tree [24,30]. SARS-CoV-1 [24,30], bovine viral diarrhea virus [33], *Klebsiella pneumoniae* [34], and *Acinetobacter baumannii* [35,36] outbreaks (Appendix A) have been studied using outbreaker and outbreaker2, available in R.

In these three methods, mutation was considered to occur during transmission, and thus, the genetic likelihood depended solely on the number of transmission events separating two individuals and not on time [24].

**Table 1 pathogens-11-00252-t001:** Epidemiological and genomic data necessary for each method. S stands for sequences, and P for phylogenetic trees. Packages are available for methods in bold. Removal time corresponds to time at which an individual becomes non-infectious, generally the culling time or end of hospitalization, and intrinsic characteristics are either number of individuals present on site or predominant animal species. Didelot et al.’s (2014) [17] method, while not based on a spatial kernel, penalized transmission trees after reconstruction if they did not respect geographical data, hence the parentheses surrounding the geographical data. Hall et al.’s (2015) [18] method could include contact data, but geographical data was used instead.

Family		Method (*Name*) [Reference]	Start of Exposure	Onset of Infectiousness	Sampling Time	Removal Time	Contact Data	Geographical Data	Intrinsic Characteristics	Phylogenetic Tree or Sequences
Non-phylogenetic		Aldrin et al., 2011 [28]		X		X		X	X	S
		**Jombart et al., 2011 (*Seqtrack*) [16]**			X					S
		Ypma et al., 2012 [32]		X		X		X		S
		**Jombart et al., 2014 (*outbreaker*) [24]**			X					S
		Worby et al., 2014 [37]			X					S
		Famulare et al. 2015 [38]			X					S
		**Worby et al., 2016 (*bitrugs*) [6]**	X		X	X				S
		**Campbell et al., 2019 (*outbreaker2*) [30]**			X		X			S
Sequential phylogenetic		Cottam et al., 2008 [2]		X	X	X				P
		Didelot et al., 2014 [17]			X			(X)		P
		Eldholm et al., 2016 [39]			X					P
		**Didelot et al., 2017** **(*Transphylo*) [40]**			X					P
		**Sashittal et al., 2020** **(*TiTUS*) [31]**	X		X	X	X			P
Simultaneous phylogenetic	Explicitly phylogenetic	Ypma et al., 2013 [5]		X	X	X		X		S
		**Hall et al., 2015** **(*beastlier*) [18]**			X	X	(X)	X		S
		**De Maio et al., 2016** **(*SCOTTI*) [41]**	X		X	X				S
		**Klinkenberg et al., 2017** **(*phybreak*) [26]**			X					S
	Implicitly phylogenetic	Morelli et al., 2012 [23]		X	X	X		X		S
		Mollentze et al., 2014 [1]			X			X		S
		Lau et al., 2015 [42]		X	X	X		X		S
		**Firestone et al., 2020 (*BORIS*) [29]**		X	X	X		X	X	S
		Montazeri et al., 2020 [43]			X					S

**Table 2 pathogens-11-00252-t002:** Modeling of unobserved processes in the non-phylogenetic family. Within-host evolution (modeled or not) includes whether the transmission bottleneck is complete or weak. When transmission is modeled, we mention the states hosts can find themselves in (S: susceptible, E: latent, I: infectious, R: removed). In addition, either geographical distance (spatial kernel), contact data, or random mixing are considered. Finally, the transmission model mentions whether there is only one index case possible (single introduction) or multiple.

Method (Name) [Reference]	Sequence Mutation	Within-Host Evolution	Transmission	Case Observation	Inference Method
Aldrin et al., 2011 [28]	Kimura model	No explicit model	SIR (infectious period)	All cases are observed but not always sampled	Partial Maximum Likelihood
		Complete	Distance kernel		
			Multiple		
Jombart et al., 2011 (Seqtrack) [16]	User’s choice	No explicit model	No explicit model	All cases are observed and sampled	Edmonds algorithm
		Complete			
Ypma et al., 2012 [32]	Deletion + Transition + Transversion	No explicit model	SEIR (latency/infectious period)	All cases are observed but not always sampled	Bayesian
		Complete	Spatial kernel		
			Single		
Jombart et al., 2014 (outbreaker) [24]	Mutation rate	No explicit model	SI (generation times)	Proportion of sampled cases	Bayesian
		Complete	Random mixing		
			Multiple		
Worby et al., 2014 [37]	Mutation rate	Pathogen population size	No explicit model	All cases are observed and sampled	Observed genetic distance vs. theoretical distribution
		Weak			
Famulare et al., 2015 [38]	Mutation rate	No explicit model	No explicit model	No assumption	Likelihood ratio test + Pruning algorithm
Worby et al., 2016 (bitrugs) [6]	No explicit model	No explicit model	SEIR (latency/infectious period)	Test sensitivity < 1	Bayesian
		No assumption	Random mixing		
			Multiple		
Campbell et al., 2019 (outbreaker2) [30]	Mutation rate	No explicit model	SI (generation times)	Proportion of sampled cases	Bayesian
		Complete	Contact data		
			Multiple		

**Table 3 pathogens-11-00252-t003:** Modeling of unobserved processes in the sequential phylogenetic family. For the sequence mutation process, NA stands for not applicable. Within-host evolution (modeled or not) includes whether the transmission bottleneck is complete or weak. When transmission is modeled, we mention the states hosts can find themselves in (S: susceptible, E: latent, I: infectious, R: removed). In addition, either geographical distance (spatial kernel), contact data, or random mixing are considered. Finally, the transmission model mentions whether there is only one index case possible (single introduction) or multiple. In the inference method, we mention how phylogenetic trees are used to infer transmission trees (either internal nodes or branches are labelled with the host or phylogenetic trees are used as a source of information). * means multiple sequences can be considered per epidemiological unit.

Method (Name) [Reference]	Sequence Mutation	Within-Host Evolution	Transmission	Case Observation	Inference Method
Cottam et al., 2008 [2]	NA	No explicit model	SEIR (latency/infectious period)	All cases are observed and sampled	Label internal nodes
		Complete	Random mixing		Maximum Likelihood
			Single		
Didelot et al., 2014 [17]	NA	Coalescent process	SIR (infectious period)	All cases are observed and sampled	Label branches
		Complete	Random mixing		Bayesian
			Single		
Eldholm et al., 2016 [39]	NA	Coalescent process	SEIR (latency/infectious period)	Probability threshold	Information source
		Complete	Random mixing		Edmonds’ algorithm
			Single		
Didelot et al., 2017 (Transphylo) [40]	NA	Coalescent process	SI (generation times)	Proportion of sampled cases	Label branches
		Complete	Random mixing		Bayesian
			Single		
Sashittal et al., 2020 (TiTUS) [31]	NA	No explicit model	No explicit model	All cases are observed and sampled	Label internal nodes
		Weak *			Logical problem

**Table 4 pathogens-11-00252-t004:** Modeling of unobserved processes in the simultaneous phylogenetic family. For the sequence mutation process, the user could either use a single substitution model or choose. Within-host evolution (modeled or not) includes whether the transmission bottleneck is complete or weak. When transmission is modeled, we mention the states hosts can find themselves in (S: susceptible, E: latent, I: infectious, R: removed). In addition, either geographical distance (spatial kernel), contact data, or random mixing are considered. Finally, the transmission model mentions whether there is only one index case possible (single introduction) or multiple. * means multiple sequences can be considered per epidemiological unit.

Method (Name) [Reference]	Sequence Mutation	Within-Host Evolution	Transmission	Case Observation	Inference Method
Ypma et al., 2013 [5]	Mutation rate	Coalescent process	SEIR (latency/infectious period)	All cases are observed and sampled	Bayesian
		Complete	Spatial kernel		
			Single		
Hall et al., 2015 (beastlier) [18]	User’s choice	Coalescent process	SEIR (latency/infectious period)	All cases are observed but not always sampled	Bayesian
		Complete *	Spatial kernel		
			Single		
De Maio et al., 2016 (SCOTTI) [41]	User’s choice	Coalescent process	Migration model	Maximum number of hosts	Bayesian
		Weak *			
Klinkenberg et al., 2017 (phybreak) [26]	Mutation rate	Coalescent process	SI (generation times)	All cases are observed but not always sampled	Bayesian
		Complete	Random mixing		
			Single		
Morelli et al., 2012 [23]	Jukes Cantor model	No explicit model	SEIR (latency/infectious period)	All cases are observed and sampled	Bayesian
		Complete	Spatial kernel		
			Single		
Mollentze et al., 2014 [1]	Kimura model	No explicit model	SEIR (latency/infectious period)	Observed cases contribute to transmission after removal time	Bayesian
		Complete	Spatial kernel		
			Multiple		
Lau et al., 2015 [42]	Kimura model	No explicit model	SEIR (latency/infectious period)	All cases are observed but not always sampled	Bayesian
		Complete	Spatial kernel		
			Multiple		
Firestone et al., 2020 (BORIS) [29]	Kimura model	No explicit model	SEIR (latency/infectious period)	All cases are observed but not always sampled	Bayesian
		Complete	Spatial kernel		
			Multiple		
Montazeri et al., 2020 [43]	Jukes Cantor model	No explicit model	No explicit model	All cases are observed and sampled	Bayesian
		Complete			

**Figure 3 pathogens-11-00252-f003:**
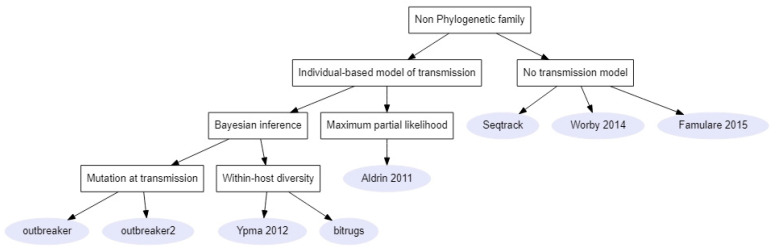
Links between methods of the non-phylogenetic family. Rectangles represent criteria on which to choose a method and the grey circles represent either the name of the method’s package or the first author and article date [28,32,37,38].

#### 2.1.2. Methods That Allow Within-Host Diversity

Worby et al. (2014) noted that previous methods based their genetic model on strong assumptions, such as a complete transmission bottleneck [24,30] or mutations occurring at time of transmission [24,30,32], thus disregarding within-host diversity. First, they constructed an approximation of the genetic distance distribution and compared it to observed genetic distances in order to determine the probability of direct methicillin-resistant *Staphylococcus aureus* (MRSA) transmission between individuals in a hospital [37]. Then, Worby et al. (2016) incorporated a genetic distance distribution approximation with an explicit transmission model tailored to a nosocomial outbreak, in a Bayesian inference framework, available in a bitrugs package in R [6]. This approach allowed for the within-host diversity previously lacking in other methods while avoiding having to make any assumptions about the within-host evolution process [6], as was necessary in their first work [37]. The transmission model considered a hospital setting, where patients were either susceptible (S) or infectious (I) one day after infection, and transmission rate per infected patient was constant until their discharge. Homogeneous mixing was assumed, meaning that each infected patient had equivalent contact with each susceptible individual. In addition, imperfect case detection was modeled by incorporating test sensibility as a model parameter [6].

#### 2.1.3. Other Methods

Conversely, in their work on infectious salmon anemia, Aldrin et al. (2011) did not use the same Bayesian approach. While they did establish a transmission model, a maximum partial likelihood approach was then used to estimate model parameters. From these estimated parameters, they calculated the probability that one salmon farm infected another. In their model, transmission probability exponentially decreased with increasing sea and genetic distances between farms and depended on farm-level characteristics such as the maximum number of fish in a cohort during the production period and when that production period was (spring vs. autumn) [28]. When genetic data was unavailable for a farm, the unknown genetic distance was imputed with the value of a parameter computed from the known genetic data [28].

Finally, the Seqtrack method [16] and Famulare et al. (2015) differ from all the others and only explicitly modeled the mutation process. Indeed, Jombart et al. (2011) computed the transmission tree for which “ancestors always precede [d] their descendants in time” (assuming sampling times follow the same chronological order as infection times) and the total genetic distance between linked nodes was minimal (i.e., the optimum branching, also named minimum spanning tree, of the graph in which all the possible links between infector and infected host are represented) using Edmonds’ algorithm [44]. While this method can be used solely with sampling times and genetic data (Table 1), other epidemiological data (e.g., locations) can also be considered to resolve equally likely ancestries. The Seqtrack algorithm was implemented in the adegenet package and has been applied to H1N1 2009 swine-origin pandemic [16], H3N8 equine influenza [45], *M. tuberculosis* [46], and *K. pneumonia* [47] outbreaks (Appendix A). Conversely, Famulare et al. identified pairs linked by direct transmission by performing a likelihood ratio test to determine whether the time of the most recent common ancestor of the considered pair (tMRCA) was equal to the earliest sampling time [38]. In order to compute the likelihood for the tMRCA, Famulare et al. assumed the mutation process followed a Poisson model with a known constant mutation rate. Competing ancestries were resolved using a pruning algorithm that the user could specify, for example, by keeping the link minimizing the time between tMCRA and sampling. This method was applied to study the Ebola virus outbreak in Sierra Leone, the 2001 H1N1 influenza pandemic, and the 2005–2008 polio outbreak in Nigeria [38] (Appendix A).

### 2.2. Phylogenetic Families

In phylogenetic families, links were established between phylogenetic and transmission trees. From the small imaginary outbreak (Figure 2), Figure 4 depicts three ways to modify the basic phylogenetic tree (Figure 2A) in order to obtain a transmission tree. Figure 4A shows a phylogenetic tree in which internal nodes are annotated with a sampled host. The transmission tree reconstructed (on the right) from this annotated phylogenetic tree contains the order of transmission but does not estimate the unknown transmission times t (unless we assume that coalescence and transmission occur at the same time). In Figure 4B, the internal nodes are annotated in the phylogenetic tree (on the left); however, the branch between two nodes hosted by different individuals is considered to be an “infection branch,” and transmission occurs along this infection branch. Therefore, we obtain a timed transmission tree (on the right) that does not assume coalescence and transmission to coincide. Finally, in 4C the possibility of annotating unobserved hosts in the phylogenetic tree is added (on the left), thus the unobserved host U can be inferred in the transmission tree (on the right).

#### 2.2.1. Sequential Phylogenetic Family

These methods (Figure 5, *n* = 5) required a phylogenetic tree to be reconstructed prior to their implementation. In one method, the phylogenetic tree was used as a source of information on the time of coalescence between two lineages [39]. Indeed, Eldholm et al. (2016) used this information in association with a SEIR model to calculate the likelihood of direct and oriented transmissions between sampled individuals of an *M. tuberculosis* outbreak [39]. When the likelihoods of transmission between every pair of individuals were calculated, the direction of transmission corresponding to the lowest likelihood was removed. Finally, the optimum branching graph was computed using Edmonds’ algorithm [44] as in Seqtrack. In order to account for unobserved cases, this method used various thresholds of direct transmission likelihoods to plot the transmission trees [39].

The four remaining methods annotated the phylogenetic tree in order to reconstruct the transmission tree using different sampling strategies of the tree space. In Cottam et al.’s (2008) method, no sampling strategy per se was implemented since the number of transmission trees compatible with their data and previous knowledge on transmission events between five farms (identified via animal movements) was relatively small (1728 trees). Every possible transmission tree was enumerated by assigning to every ancestral node one of its two descendants, while moving backwards in time on the phylogeny [2] (node annotation similar to Figure 4A, with added constraints). Then, the likelihood of a transmission tree was computed from the joint likelihood of each transmission pair, which was based on the probability of the epidemiological data (removal dates and onset of infectiousness, Table 1) according to the SEIR transmission model (Table 3). This method was applied to the 2001 FMD outbreak in the United Kingdom (Appendix A).

Similarly, Sashittal and El-Kabir (2020) aimed to label the internal nodes in a phylogenetic tree reconstructed from HIV sequences [31] (node annotation similar to Figure 4A). However, in this method, a weak transmission bottleneck was considered. Moreover, the labelling was not restricted to the two descendants of each node. While the transmission process was not explicitly modeled (Table 3), the labelling had to satisfy a number of constraints derived from the known transmission windows (i.e., from exposure time to removal time) and contact information (Table 1). The transmission tree reconstruction was treated as a logical problem and a parsimonious consensus tree was then selected from uniformly sampled transmission trees that satisfied the temporal and contact constraints [31].

Methods that identify transmission events as branching events in a phylogeny and assume a complete bottleneck do not consider within-host evolution [17]. Thus, Didelot et al. (2014) inferred the transmission tree by affecting hosts along branches in the phylogenetic tree [17,40] (branch annotation similar to Figure 4B). Since hosts could change along branches and not only at the nodes, transmission events were no longer restrained to the timing of coalescent events. In their method, Bayesian inference was used to infer the epidemiological parameters of their SIR (Susceptible-Infected-Removed) model, the within-host evolutionary parameters (for which they considered a neutral coalescent process with constant population size Ne and average population generation time g, i.e., duration of the replication cycle), and the transmission tree. Thus, contrary to the complete enumeration in Cottam et al.’s (2008) method and the uniform sampling used in Sashittal and El-Kabir (2020), MCMC (Markov Chain Monte Carlo) sampling was used to explore the transmission tree space. In addition, they used geographical data as well as diagnostic test results to penalize transmission trees [17].

The main limitation of previous methods is the assumption that the outbreak is finished and that all cases were sampled [40]. Didelot et al. (2017) therefore implemented another Bayesian method in an R package called Transphylo, where the user could define the probability for an individual to be sampled and either select the completed or the ongoing outbreak scenario (branch annotation and unobserved hosts similar to Figure 4C). Contrasting with their previous work, the transmission model considered was a branching process [40]. The branching process was defined by a number of offspring distribution (i.e., number of individuals one individual can infect) and a generation time distribution [40]. The Transphylo package was chosen to study (Appendix A) bacterial transmission (such as *M. tuberculosis* [48,49,50] and *K. pneumoniae* outbreaks [51,52]), as well as viral transmission (e.g., part of the recent SARS-CoV-2 pandemic [53] and a large mumps outbreak in Canada [54]). Recently, the Transphylo package [40] was extended to infer transmission trees from multiple phylogenetic trees [55].

None of these transmission tree reconstruction methods explicitly modeled sequence mutation since the method is applied to an already fully reconstructed phylogenetic tree (hence the “not applicable” in Table 3). However, some articles [39,40,48,51,53,54,55] have used substitution models to reconstruct the phylogenetic tree prior to the implementation of their method.

#### 2.2.2. Simultaneous Phylogenetic Family

Five methods from this family (Figure 6) implicitly considered a phylogenetic tree where internal nodes corresponded to transmission events (node annotation similar to Figure 4A). Morelli et al. (2012) built a likelihood function taking into account correlations between genetic and epidemiological data to study the 2001 and 2007 FMD outbreaks in the United Kingdom [23]. Indeed, the genetic pseudo-likelihood depended on the time from infection to observation and therefore indirectly permitted mutations to occur within the host without explicitly modeling within-host evolution. The transmission model was then extended by Mollentze et al. (2014) to allow multiple introductions of the disease instead of a single index case, which is more suited to endemic situations, and was applied to a canine rabies outbreak in South Africa [1] (Appendix A). Both works otherwise used a similar SEIR transmission model (Table 4) and estimated parameters including time-to-infectiousness (or latency period) and time-to-sampling distributions [1,23]. However, Mollentze et al. (2014) indirectly modeled unobserved cases by allowing observed cases to transmit after their removal time and considered two categories of individuals, those that could transmit the virus (dogs) and those that could not [1].

Lau et al. (2015) noted that these previous methods lacked a way to explicitly infer the unobserved transmitted sequences. Indeed, Morelli et al. (2012) and Mollentze et al. (2014) considered a genetic pseudo-likelihood computed for only observed sequences [1,23]. Therefore, Lau et al. (2015) proposed a genuine joint inference by modeling missing genetic data and inferring the unobserved sequences alongside the transmission tree [42]. In their transmission model, two types of infections were considered: primary infections corresponding to imported cases whose sequences were derived from a universal sequence G_M_ and secondary infections. Secondary infections were modeled according to a SEIR model [42]. Hayama et al. (2019) applied this method to the 2010 FMD outbreak in Japan [56] (Appendix A). BORIS is an extension of Lau et al.’s model that incorporates farm-level covariates, such as the number of animals and predominant species (Table 1), which are considered to influence susceptibility and infectiousness of farms in the transmission model [29].

The most recent method from this sub-category did not take into account an explicit transmission model [43]. Montazeri et al. (2020) provided two algorithms that reconstructed the phylogenetic tree from a possible transmission tree by considering estimates of infection times and the absence of within-host diversity. Montazeri et al. applied this method to an HIV transmission cluster in San Diego, California, and the 2014 Ebola virus outbreak in Sierra Leone.

Contrary to these five previous methods, four methods aimed to simultaneously infer phylogenetic and transmission trees. In these four Bayesian methods, a formal link is established between phylogenies and transmission trees and in each MCMC step, both trees are updated in a way that guarantees they remain compatible. Ypma et al. (2013) considered a hierarchical tree where every within-host phylogeny was connected through transmission [26]. They focused on a previously studied 2001 FMD outbreak [2,23] and assumed all infected individuals were known [5] (Table 4). Similarly, Klinkenberg et al.’s (2017) method [26] considered a hierarchical tree. This method was implemented in the R package phybreak and was applied to five published datasets: *M. tuberculosis* [17], MRSA, two FMD outbreaks [2,5,23], and H7N7 [18] (Appendix A).

Instead of individually modifying within-host phylogenies as in the hierarchical tree approach, Hall et al.’s (2015) method partitioned the phylogeny by annotating internal nodes with hosts then estimating a parameter for each host to determine their time of infection along the branch (branch annotation similar to Figure 4B) [18]. Hall et al. (2015) studied a 2003 H7N7 outbreak at a farm-level and divided avian farms into two categories (“high-risk” vs. “low-risk”), which differed in the distribution of their infectious period due to the implementation of control measures [18]. Case observation was not modeled, while missing genetic data was replaced by non-informative sequences (repetition of nucleotide “N”) [18]. This method implemented in the beastlier package in BEAST [18] was then applied to a H5N8 avian influenza outbreak (Appendix A) with birds as epidemiological units [57].

In these three methods, the transmission process was modeled by epidemiological models previously mentioned in the literature, such as a homogeneous branching process [26] or an individual-based SEIR model, which included a function describing host characteristics affecting transmission, such as geographical distances (via a spatial kernel) [5,18] or risk group [18]. However, De Maio et al. (2016) [41] had an original approach and used the Bayesian structured coalescent approximation (BASTA, [58]). They considered hosts as separate populations characterized by their exposure interval (time from start of exposure to removal, Table 1) and between which pathogens can migrate. This transmission model allowed multiple infections of the same host and transmission of multiple strains during an infection. Therefore, this method implemented in the SCOTTI package [41] in BEAST2 [14] was more suited to outbreaks with frequent mixed infections and large transmission inocula and was applied to FMDV and *K. pneumoniae* outbreaks (Appendix A). In addition, this method is the only one in this family (Table 4) that modeled the case observation process (the user could specify a maximum number of hosts in the outbreak) (branch annotation and unobserved hosts similar to Figure 4C).

All these methods explicitly modeled the sequence mutation process with a substitution model, and four out of nine modeled the within-host evolution with a coalescent process (Table 4).

### 2.3. Application to M. tuberculosis, FMDV, and MRSA Outbreaks

*M. tuberculosis* is characterized by a low mutation rate (Appendix A) coupled with a high proportion of identical sampled sequences [21]. Infection by *M. tuberculosis* can lead to a long latency period, and the majority of cases are asymptomatic. Thus, we should not assume that all cases are observed, and not accounting for the possible long latency could lead to incorrect transmission tree inference. However, the within-host evolution could be disregarded considering the low mutation rate. Methods (included in a package) that allow imperfect case detection are outbreaker and outbreaker2, bitrugs, Transphylo, and SCOTTI (Table 2 and Table 4). Among these five methods, Transphylo, outbreaker, and outbreaker2 could allow for a long latency period by selecting an appropriate generation time distribution (Appendix A), as has previously been demonstrated with the Gamma generation time density in Transphylo [40]). *M. tuberculosis* outbreaks have been reconstructed using five methods from phylogenetic and non-phylogenetic families: Seqtrack (NPF) [46], Didelot et al. (2014) [17], Eldholm et al. (2016) [39], and Transphylo [40,48,49,50,55] from the SeqPF and phybreak (SimPF) [26] (Appendix A).

Conversely, FMDV has a high mutation rate (Appendix A), and farms are generally the most relevant epidemiological units in an FMDV outbreak. In addition, wind-mediated transmission can play a role in disease spread [3], and pigs shed more than ruminants, who are more susceptible to FMDV [59]. Thus, disregarding within-host evolution seems difficult to justify when the “host” is a farm and the pathogen has a high mutation rate. Moreover, considering the fact that farms have fixed locations and the role played by indirect transmission, it seems unwise to assume random mixing of hosts as well as disregard the information provided by geographical data. Finally, considering the predominant species in the transmission model could help exploit the dissymmetry in roles played by pig and cattle farms. The methods (included in a package) that have an individual-based transmission model with a spatial kernel are BORIS and beastlier (Table 4). However, while BORIS takes into account farm characteristics, beastlier models within-host evolution (Table 4). Seven methods have been applied to FMDV outbreaks: Cottam et al. (2008) (SeqPF) [2], Ypma et al. (2013) [5], SCOTTI [41], phybreak [26], Morelli et al. (2012) [23], Lau et al. (2015) [42,56], and BORIS [29] from the SimPF (Appendix A).

MRSA has a low mutation rate (Appendix A); however, within-host diversity is important to consider when studying *S. aureus* [40]. Studied outbreaks have taken place in neonatal ICUs [6,30,60]. A hospital setting implies a higher proportion of sampled or at least detected cases and multiple possible introductions. Detailed contact data could be available. Therefore, methods used to reconstruct a MRSA outbreak could assume that all cases are observed. However, depending on the outbreak, assuming a single disease introduction could be inappropriate. In addition, contact data would be interesting to consider. Methods (included in a package) that do not assume a single disease introduction are Seqtrack, outbreaker and outbreaker2, bitrugs, and BORIS (Table 2 and Table 4). Among these, only bitrugs allows within-host diversity and was specifically designed to study a nosocomial outbreak, while outbreaker2 considers contact data (Table 1). Therefore, the choice between the two methods depends on the type of data available and whether accounting for within-host evolution is necessary to answer our question about the studied outbreak. Bitrugs [6,60] and phybreak [26] were chosen to study MRSA outbreaks in neonatal ICUs (Appendix A).

## 3. Discussion

We systematically reviewed the literature for methods combining genomic and epidemiological data to reconstruct transmission trees. The epidemiological data necessary to implement each method was first used to differentiate them. Methods were then divided into three families according to the way genetic data was integrated in the transmission tree inference. We thus differentiated the methods in order to offer practical considerations to examine when selecting transmission tree reconstruction methods.

We were interested in the integration of epidemiological and genetic data in transmission tree inference; however, two methods (Cottam et al. 2008 and Seqtrack) [2,16] were criticized by others for not fully integrating the information provided by both types of data. Even though the possible transmission trees were based on the phylogenetic tree, Cottam et al. (2008) [2] calculated transmission tree likelihood solely from epidemiological data, disregarding any further information that could have been derived from the genetic data [32]. Similarly, Seqtrack [16] only considered additional epidemiological data to distinguish multiple cases when their genetic sequences were identical [32].

The non-phylogenetic family estimated transmission probability from calculated pairwise genetic distances. However, two families used phylogenetic trees to reconstruct transmission trees, either by inferring the host of each node or branch in the phylogenetic tree [2,17,18,31,40,41], considering within-host phylogenetic trees as part of a hierarchical tree [5,26], or by using the phylogenetic tree as a source of information [39]. In the sequential phylogenetic family, phylogenetic trees were reconstructed prior to the implementation of the method and thus called for an additional choice, the phylogenetic tree reconstruction method. Moreover, the phylogenetic tree needs to be correctly reconstructed, or it will lead to errors in the transmission tree. At first, all sequential phylogenetic methods used a single fixed tree generated beforehand by a standard phylogenetic method as an input. As such, these methods ignored any uncertainty in the estimation of the phylogeny [18] and therefore did not take the full uncertainty in the evolutionary process into account [26]. Thus, the Transphylo package was extended to reconstruct transmission trees from multiple phylogenetic trees [55]. However, another strategy was to infer transmission trees and phylogenetic trees simultaneously; we grouped these methods in the simultaneous phylogenetic family.

As mentioned by Klinkenberg et al. (2017), four unobserved processes could be taken into account or ignored [26]: sequence mutation, within-host evolution, transmission, and case observation. Substitution models explicitly model sequence mutation, while genetic distances calculated without a substitution model do not consider intermediary or back mutations and can therefore lead to incorrect estimates. Sequential phylogenetic methods either modeled the mutation process indirectly or did not model it, depending on the method used to pre-generate the phylogenetic tree. Cottam et al. (2008) used a parsimony method [2], while the others [17,39,40] generally opted for Bayesian methods, which supported a number of substitution models. In the two remaining families (non-phylogenetic family and simultaneous phylogenetic family), all methods had the similar option to take into account an explicit substitution model.

Since we expect a non-negligible within-host evolution in infections by pathogens with long generation times [17] combined with a high evolutionary rate, ignoring the fact that mutations occur within-host (e.g., by considering mutations that occur at transmission, such as in outbreaker [18,26]) is inappropriate in this case. In addition, some methods (Morelli et al. 2012, Mollentze et al. 2014, and Lau et al. 2015), while allowing for within-host mutation, only allowed a single pathogen lineage to exist within each host at any given time [18], therefore disregarding any within-host diversity. However, when dealing with a highly sampled outbreak, Ypma et al. stated in 2013 that ignoring within-host diversity’s contribution to the observed differences between sampled sequences could lead to incorrect inference of the transmission tree [5]. Methods that modeled within-host evolution generally assimilated it into a coalescent process [5,17,18,26,39,40,41], which requires the assumption of a low sampling fraction within the host [18]. While this condition is usually verified at an individual scale, it should be kept in mind when reconstructing an outbreak between farms, where the “host” is actually a group of individuals.

Furthermore, farms as epidemiological units could also make it more difficult to disregard within-host population diversity and assume a single infection (multiple introductions are likely to occur), as well as a single within-host pathogen lineage. The reconstructed transmission trees generally considered only the first transmission event, or when it was necessary to account for these secondary transmission events, hosts could simply be duplicated in the transmission tree and infection events were considered independent [24]. Aldrin et al. disregarded completely the possibility of multiple infections of the same farm and chose the least distant genetic data when multiple sequences were available for one farm [28]. The possibility of transmitting genetically diverse strains was overlooked in most methods due to a strong assumption, that is, a transmission bottleneck size of one transmitted sequence [37]. This assumption was relaxed in three methods, Worby et al. (2014), for whom transmission bottleneck size varied [37], and De Maio et al. (2016) and Sashittal et al. (2020), who disregarded transmission bottlenecks completely, allowing the transmission of multiple strains [31,41] and even multiple infections in SCOTTI [41].

While epidemiological models contribute to estimating the most probable transmission tree, a number of underlying assumptions are made on the natural history of the disease and how the disease spread, which need to be considered before choosing a method. For instance, assuming random mixing between hosts means that every infected host is equally likely to infect any susceptible host (used in Didelot et al. 2014, Eldholm et al. 2016, and bitrugs) [6,17,39]. This could be problematic, for example, when considering an FMD outbreak between farms where wind-mediated transmission can play a role in disease spread [3], and thus transmission between farms is no longer equally likely but depends on wind direction and geographical distances. Therefore, some methods have used an individual-based model with a spatial kernel [1,5,18,23,42] or even included farm characteristics influencing infectivity and susceptibility, such as predominant species or herd size (BORIS) [29]. Lastly, considering that the outbreak has a single introduction event is not suited to an endemic situation [1] or even the spread of nosocomial infections in a hospital setting, where multiple introductions can occur [6]. Therefore, some methods did not assume a single disease introduction and either identified genetic outliers [1,24,30] or included disease introduction in the transmission model [6,29,42]. Five methods (Seqtrack, Worby et al. 2014, Famulare et al. 2015, Montazeri et al. 2020, and TiTUS) did not explicitly model transmission.

The final unobserved process to be considered is case observation. According to Didelot et al. (2017), the main limitation of some works preceding the development of Transphylo was the assumption that the outbreak was over and that all cases had been sampled [40]. Indeed, assuming all cases to be linked by direct transmission leads to incorrect estimates on the natural history of the disease or false transmission links. Thus, some methods explicitly modeled case observation by estimating a proportion of observed cases [24,30,40], test sensitivity [6], or the maximum number of hosts in the outbreak [41]. Mollentze et al. (2014) indirectly accounted for unobserved cases by allowing hosts to transmit the pathogen after their removal [1]. Whether the case observation process needs to be modeled in a transmission tree reconstruction method depends on the possibility of missing infected individuals in the studied outbreak. Therefore, natural disease history, testing strategies, and their effectiveness should be considered.

Moreover, the choice of a method also depends on its availability, as well as its applicability to a wide range of datasets. This can be attested by the number of studies found in our search that reconstructed transmission trees with methods available in packages (e.g., Seqtrack algorithm, *n* = 4 [16,45,46,47], outbreaker, *n* = 4 [24,34,35,36], and especially Transphylo, *n* = 9 [40,48,49,50,51,52,53,54,55]) compared to methods like Ypma et al. (2013) and Morelli et al. (2012) [5,23], which are designed for specific datasets and rarely used for other purposes. Unfortunately, computational time was not always available in the selected articles, which makes it difficult to estimate the size of the dataset that can be studied.

Finally, we decided to exclude methods that needed deep-sequencing data. For instance, a Bayesian inference method called BadTrIP (BAyesian epiDemiological TRansmission Inference from Polymorphisms) using genetic and epidemiological data considered a genetic data format (in the form of nucleotide counts for each position in the genome) [61] that greatly differed from the other methods. Another method called SLAFEEL (Statistical Learning Approach For Estimating Epidemiological Links) considered a set of sequences for each host, and epidemiological data was used to calibrate a penalization of the pseudo-likelihood (describing the probability of obtaining the set of sequences in the infected host from the set of sequences present in the infector) [62]. These methods (which do not constitute an exhaustive list) could be interesting to use when multiple sequences are available for a host, when usual model assumptions are unsuitable (SLAFEEL), or when we cannot assume the absence of recombination (BadTrIP).

The choice of a transmission tree reconstruction method thus depends on the characteristics of the pathogen such as mutation rate and natural history of the disease, the epidemiological and genetic data available from the outbreak, as well as the questions we wish to see answered. The impact that violating underlying assumptions of the evolutionary and epidemiological models has on the reconstructed transmission tree, as well as the use of biased data, would be interesting to further investigate.

## 4. Materials and Methods

### 4.1. Search Strategy

We searched two electronic databases, Pubmed and Scopus, from 13 October to 17 November 2020. The list of references from the selected studies were screened in order to find further studies to be included. We selected keywords revolving around transmission trees (“transmission chain”, “transmission tree”, “transmission reconstruction”, “transmission network”, “who infected whom”) and those pertaining to the use of genomic data (“genome”, “SNP”, “genetic data”, “phylogenetic data”). We formulated the following search query: (“transmission chain” OR “transmission tree” OR “transmission reconstruction” OR “transmission network” OR “who infected whom”) AND (“genome” OR “genomic” OR “sequence data” OR “genetic data” OR “phylo* data”). Depending on search databases, the search query was entered in “all fields” (Pubmed) or in “Title, abstract, or author-specified keywords” (Scopus). In the database that did not support wild cards (Scopus), “phylo* data” was replaced by “phylogenetic data”.

### 4.2. Eligibility Criteria

Studies were included when they inferred a transmission tree for an infectious disease outbreak using non-simulated epidemiological and genomic data. The genomic data considered was single-nucleotide polymorphisms identified from consensus sequences or the consensus sequences of entire genes themselves and not deep sequencing data, where multiple nucleotides are available for a single locus. We defined a transmission tree as a rooted graph consisting of nodes (representing cases, i.e., infected individuals or groups of individuals) connected by edges (representing transmission events). Transmission trees reconstructed using solely one type of data were excluded. Methods that estimated possible transmission events compatible with the epidemiological data separately from those compatible with the genomic data were excluded. Even if they graphically combined these transmission events or compared the results obtained by each type of data, in the absence of an algorithm linking the two types of data to reconstruct a transmission tree, we considered them to not formally combine epidemiological and genomic data.

### 4.3. Data Management

Citations were exported from the two electronic databases to EndNote X9 (2018), where we proceeded to remove duplicates and screened the title, abstract, and when necessary to reach a decision, the material and methods section of the remaining articles. The full texts of selected articles were then assessed for eligibility in chronological order to better understand how the methods relate to one another and their interdependency.

### 4.4. Data Collection Process

We recorded the inference method (e.g., Bayesian, maximum-likelihood) and the limits of a reconstruction method when they were discussed in an article.

Since genetic diversity affects the ability to reconstruct transmission histories [20], we systematically sought the following information concerning the genomic data. We documented the pathogen, the mutation rate, the number of genetic sequences, and the number of single-nucleotide polymorphisms or the sequence length used to reconstruct the trees, as well as the time period covered. When pathogen mutation rate was not estimated in the article, we searched the literature for this information.

We recorded the epidemiological unit studied, for example, individual or group of individuals. We sought this information because depending on the epidemiological unit, within-host evolution can mean either intra-individual pathogen evolution or intra-group, and therefore incorporate transmission dynamics between individuals within the group considered as a host. Moreover, we identified the type of epidemiological data needed and recorded computational time when available, in order to give practical reasons for method selection. Types of epidemiological data included start of exposure, onset of infectiousness, sampling time, removal time, contact and geographical data, as well as intrinsic characteristics that could influence either infectiousness or susceptibility. For instance, predominant species are intrinsic characteristics of a farm that could be interesting to include in the transmission model of an FMD outbreak [29]. Indeed, pigs shed more virus than ruminants, who are more susceptible; therefore, the most likely pattern of airborne FMDV spread is from pig to cattle and sheep [59].

Finally, we were interested in whether unobserved processes (e.g., mutation, within-host evolution, transmission, and case observation) were explicitly modeled.

Substitution models (e.g., Kimura [63] and Jukes Cantor [64] models) are often used to describe sequence mutation. We recorded the type of substitution model used for the sequence mutation.Within-host evolution can be modeled by population models (e.g., the coalescent [65]) that are commonly used in phylogenetic tree reconstruction to describe the ancestry between sampled pathogens. When possible, we recorded the population model describing the within-host evolution.Three sub-categories were considered to describe the transmission model. Since an individual’s infectiousness varies over time depending on pathogen shedding [66], transmission models consider different stages of an infectious disease according to transmission potential. Parameters such as latency period and generation time can be fixed beforehand or estimated in the inference. The latency period corresponds to the time from infection by a pathogen to onset of infectiousness and is followed by an infectious period during which the individual can transmit the pathogen to others [67]. Generation times (Tg) represent the time interval between the infection of an index case and the time of transmission from that index case to secondary cases; Tg are related to the latency and infectious periods but also to the variation of an individual’s infectiousness over time [68]. Thus, we identified the different states considered for a host (for instance, S: susceptible, E: exposed, I: infectious, R: removed) and whether latency and infectious periods or generation times were considered to model the natural history of the disease. Moreover, since a transmission event is the result of direct or indirect contact between an infectious individual and a susceptible individual, this contact can be modeled by assuming a random mixing of individuals, considering transmission probability as a function of geographical distances (i.e., a spatial transmission kernel) or taking into account explicit contact data. In our second subcategory, we were interested in how contacts between hosts were modeled (random mixing, spatial kernel, or contact data). Finally, we recorded whether the method assumed that a single introduction of the disease was responsible for the outbreak or if multiple introductions into the host population were possible.For case observation, we were interested in how the methods accounted for imperfect case detection and whether all observed cases were sampled or if the method had a way to handle missing genomic data.

## Figures and Tables

**Figure 1 pathogens-11-00252-f001:**
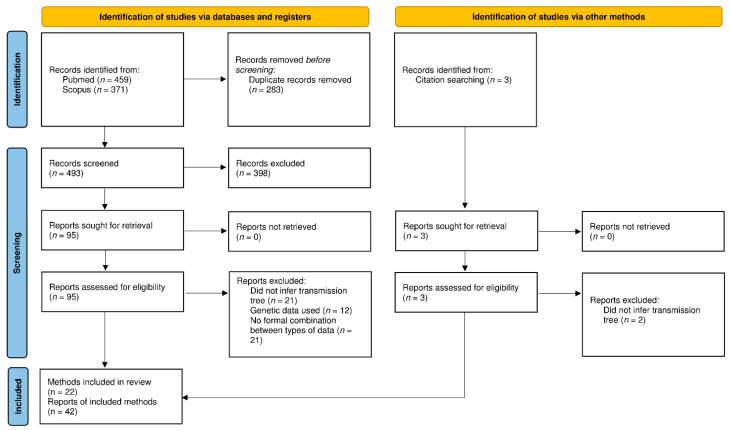
PRISMA flow diagram representing the article selection process (from [27]).

**Figure 2 pathogens-11-00252-f002:**
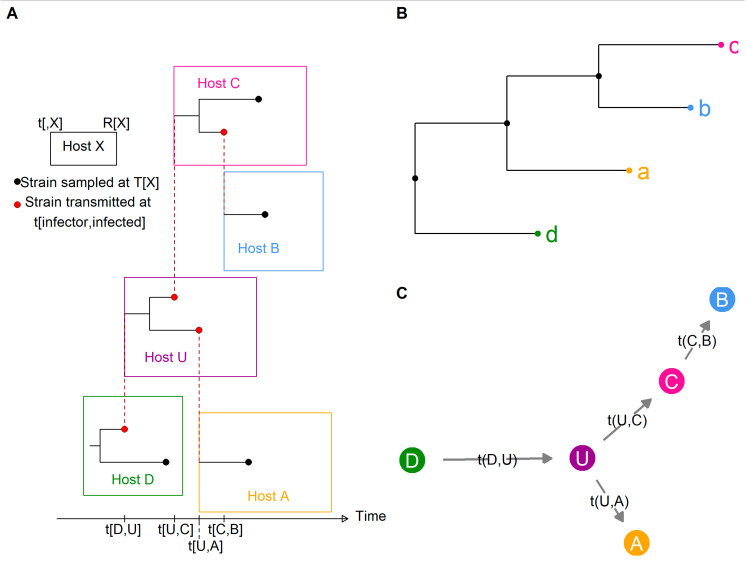
A simple transmission scenario (**A**), the reconstructed phylogenetic tree (**B**), and the transmission tree (**C**). Rectangles represent hosts, and black lines within a rectangle represent within-host evolution of the pathogen. Black circles correspond to sampled strains, red circles to transmitted strains, and red dotted lines to a transmission event. Length of host rectangles represent time from infection (t) to removal (R). The phylogenetic tree is reconstructed from sequences (a, b, c, and d) sampled at time T. The transmission tree considered the unobserved host U.

**Figure 4 pathogens-11-00252-f004:**
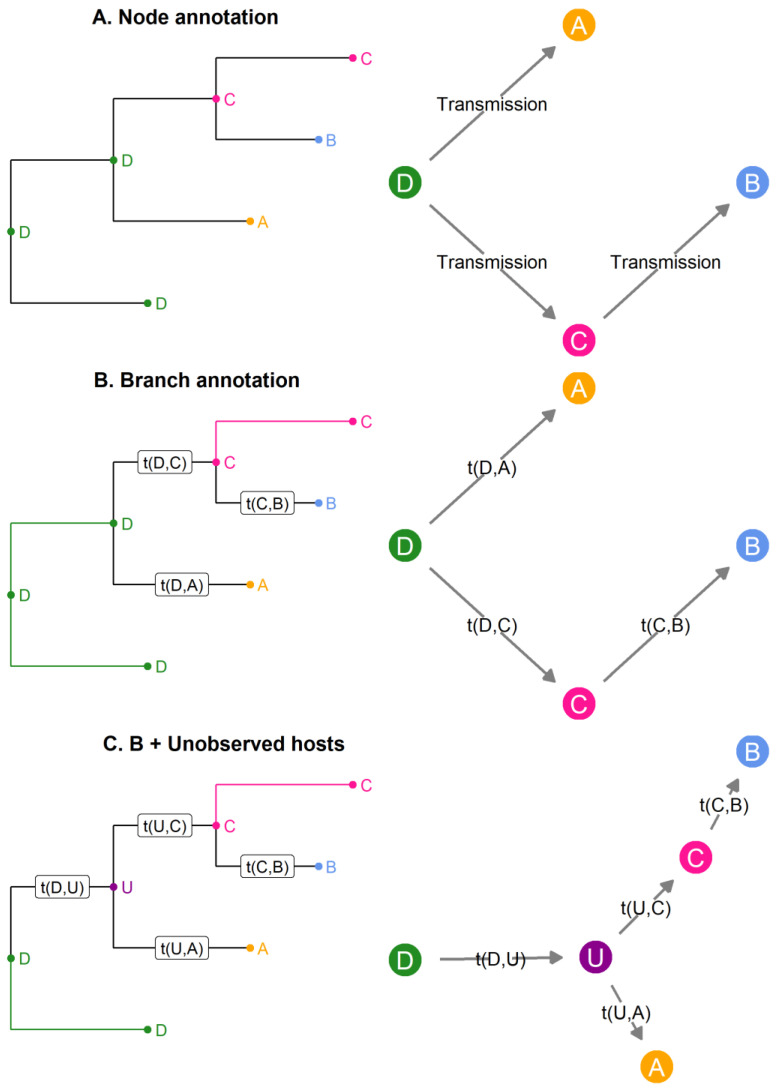
Three links between phylogenetic (on the **left**) and transmission trees (on the **right**). Node annotation with observed hosts (**A**) leads to the identification of transmission links. Annotating the branches (**B**) adds on the time of transmission t. Annotating branches with observed and unobserved hosts (**C**) means the identification of host U is possible.

**Figure 5 pathogens-11-00252-f005:**
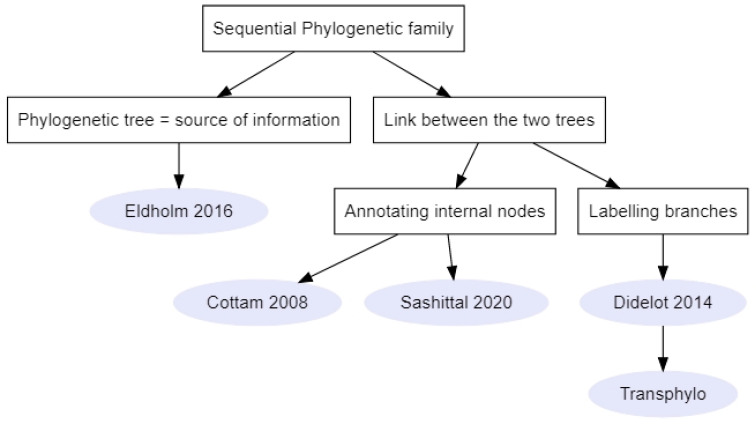
Links between methods of the sequential phylogenetic family. Rectangles represent criteria on which to choose a method and the grey circles represent either the name of the method’s package or the first author and article date [2,17,31,39].

**Figure 6 pathogens-11-00252-f006:**
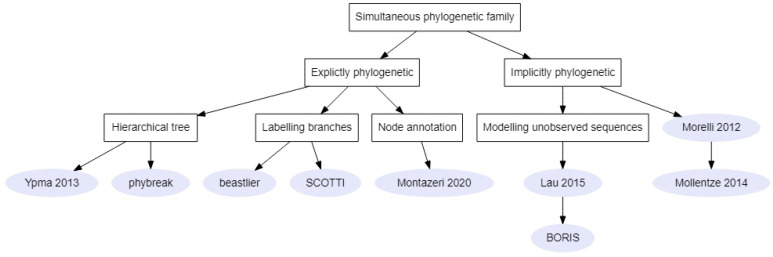
Links between methods of the simultaneous phylogenetic family. Rectangles represent criteria on which to choose a method and the grey circles represent either the name of the method’s package or the first author and article date [1,5,23,42,43].

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
