# Peer review of "Methods Combining Genomic and Epidemiological Data in the Reconstruction of Transmission Trees: A Systematic Review"

_pathogens, 2022, doi:10.3390/pathogens11020252_

Round 1
Reviewer 1 Report
This is a systematic review of methods for the reconstruction of pathogen transmission trees using genomic data. It is, in totality, a very strong piece of work that has the potential to be a comprehensive reference source for these methods and is a valuable contribution to the field. The manuscript is thoughtfully laid out and easy to follow (I particularly like figures 2-4). There are three areas where I have slight reservations with this iteration of the work.
I will unblind myself to take issue with the point in lines 485-495, as I also raised this at the Epidemics conference. I would say that Morelli et al., Mollentze et al., Lau et al., and Firestone et al. (BORIS) all belong with Montazeri et al. in the simultaneous phylogenetic family. I’d make two arguments for this.
Firstly, there is the simple existence of Montazeri et al. That paper proposes a tree in which internal nodes represent transmissions and mutation happens along branches, exactly as Morelli et al. and its derivatives do. The only real difference is that Montazeri et al. describe their method as being phylogenetic while the others do not. And as such, I think it is difficult to support the idea that they belong in entirely different categories. It would be quite easy to implement the Morelli model in e.g. BEAST, as the object it works with is basically identical to a dated phylogeny. Secondly, my previous attempt at a classification can be found in Hall et al., doi:10.20506/rst.35.1.2433 (I do not give that reference to insist that you defer to its content, rather just to avoid repeating it.) I separated the phylogenetic class into those that assume internal nodes are transmissions, and those that do not. Note that the former assumption is actually common in phylodynamics (e.g. Stadler et al., doi:10.1093/molbev/msr217, Volz et al., doi:10.1534/genetics.109.106021) where it is generally seen as reasonable under many circumstances. Thus the idea that an internal node represents a common ancestor rather than a transmission (L57-60) is not universal in self-declared phylogenetic methods, and making some other assumption does not make a method non-phylogenetic. If those phylodynamic methods qualify as phylogenetic (and I would not like to argue that they do not), then Morelli et al. does.
There are a few more papers that arguably have been missed here – e.g. Numminen et al., doi:10.1098/rspb.2014.1324 and Famulare and Hu, doi:10.1093/inthealth/ihv012, although it is possible that these were found and discarded due to the exclusion criteria. (The authors might want to consider listing what the 49 excluded papers were.) But if they were indeed missed, they probably need to be added, and there may be others out there. The existing Hall et al. review (see above) as well as Firestone et al., doi: 10.1038/s41598-019-41103-6 would be good starting points to check. With even five of the included papers being “identified from scanning references and other sources” (the nature of the “other sources” is not given in the text), it would seem that the database search strategy lacked recall. While I do not insist on this, it might be useful to go back and refine it so that all the missing ones were found; this might be a very useful tool when the time comes for an update to this paper.
My final major comment is that while I do not argue with this paper’s decision to not explore deep sequencing methods, the term “deep sequencing” is not defined and the boundaries of this class are not established. Plenty of the methods described here allow for multiple samples per epidemiological unit and in that case you could use those kinds of data with them (at least in theory; there would probably be computational issues with such big datasets in practice). Maybe a more rigorous place to draw the line is to restrict to those that can be used on consensus sequences alone. And while this is not the place to rattle off (and cite) an exhaustive list of deep sequencing methods, the discussion paragraph could do with mentioning that SLAFEEL and BadTrIP are by no means an exhaustive list.
But in conclusion, I reiterate that I really like this paper and congratulate the authors on putting it together.
Minor comments:
L36: “Transmission events generally correspond to the first transmission event” – please reword this?
L95: Insertions and deletions are hard to distinguish from each other, hence the term “indel” is generally used. (Also I’m not sure any of these methods are set up to deal with indels.)
L97: “of the same nature” is vague; this paper probably does not need to go into the transition/transversion distinction anyway.
Tables 1/3: “Eldholm” is misspelled “Eldoholm”.
Table 2: The vertical alignment of entries in columns 3 and 4 is a bit confusing – maybe switch the table to landscape and rotate 90 degrees if there are space issues?
Also table 2: I understand entirely what you mean, but the difference between “observation” and “sampling” is not explicit in the text.
L325: Exploring every possible transmission tree that is compatible with the phylogeny would be ideal, but the trouble is that this space is too big in general; perhaps note here that the reason Cottam et al. could do this is that their tree was small.
L344: Sashittal and El-Kabir definitely do consider within-host diversity; anything that allows an incomplete bottleneck must by definition as it allows diversity to be transmitted.
L498-505: There is a slight contradiction here, as you start by saying all these methods all start with a single fixed tree and then shortly afterwards report that the new TransPhylo does not.
Reviewer 2 Report
The review done in reconstruction of transmission tree methods is relevant and in my opinion is a good starting point to help selecting the appropriate method for an outbreak.
Nevertheless, the phylogenetic tree reconstruction is insufficiently depicted and references are almost only related to bayesian methods. In my opinion, the authors should refer other kinds of methods such as the distance-based phylogenetic inference methods (consider for instance this review, "Distance-based phylogenetic inference from typing data: a unifying view, Briefings in Bioinformatics, 2021") and some tools in this context that also integrate sequence data with epidemiological data, such as "PHYLOViZ: phylogenetic inference and data visualization for sequence based typing methods, BMC bioinformatics, 2012"
Notice also when, in page two you refer the number of nodes, it misses you a reference that also take into account the number of arcs, namely "Not Seeing the Forest for the Trees: Size of the Minimum Spanning Trees (MSTs) Forest and Branch Significance in MST-Based Phylogenetic Analysis, Plos One 2015"
